# Microsporidiosis Causing Necrotic Changes in the Honeybee Intestine

**Aneta A. Ptaszyńska** [1,*] and **Marek Gancarz** [2,3]

1. Department of Immunobiology, Institute of Biological Sciences, Faculty of Biology and Biotechnology, Maria Curie-Skłodowska University, Akademicka 19 Str., 20-033 Lublin, Poland
2. Faculty of Production and Power Engineering, University of Agriculture in Krakow, Balicka 116B, 30-149 Krakow, Poland
3. Institute of Agrophysics, Polish Academy of Sciences, Doświadczalna 4, 20-290 Lublin, Poland
* Correspondence: aneta.ptaszynska@mail.umcs.pl

**Featured Application: Microsporidiosis leads to the formation of dead intestine cell spots, which can cause interruption of the host's intestinal continuity and lead to intestinal leaking.**

**Abstract:** Background: Microsporidia from the *Nosema* (*Vairimorpha*) genus are pathogenic fungi that complete their life cycle in the honeybee intestine. Therefore, the aim of this study was to determine the impact of the course of infection on the viability of honeybee intestine cells. Methods and Results: Intestines isolated from healthy and *N. ceranae*-infected honeybees were stained using two dyes, SYTO 9 and propidium iodide, and analyzed under an Axiovert 200M fluorescence microscope immediately after the isolation of the intestines. The ImageJ program was used for the quantitative analysis of the cell structure parameters. Our study demonstrated for the first time that healthy bees have a higher number of live cells in their intestines than infected bees, and that the intestines of *N. ceranae*-infected honeybees contain dead cells concentrated in spots. The results obtained for these two cases differed significantly, and were confirmed by statistical tests. Conclusions: The intestines of infected honeybees contain dead cells concentrated in red/dead spots, which can lead to necrotic changes, the interruption of the host's intestinal continuity, intestinal leaking and the increased mortality of the host.

**Keywords:** nosemosis; pollinators; bee; intestinal leaking; live/dead tests; midgut; PI; Syto 9

## 1. Introduction

Microsporidia are intracellular pathogens that belong to the Fungi kingdom [1]. Microsporidia can cause severe infections in animals, including humans, that are particularly severe in immunocompetent and immunodeficient hosts; this includes patients undergoing organ transplant, patients with advanced human immunodeficiency virus (HIV) infection, patients receiving immune modulatory therapy, etc. [2]. The life-threatening infections that are caused by microsporidia motivate the intensive research of this group of microorganisms. As such, the taxonomy of microsporidia is in a state of flux. Economically and environmentally important insects such as honeybees can be infected by three species from the Nosematidae family, i.e., *Nosema apis*, *N. cearanae* and *N. neumanii* [3–5]. The taxonomy of the Nosematidae family was revised by Tokarev et al. [6], who postulated that the *Nosema* genus should be incorporated into the *Vairimorpha* genus in one monophyletic lineage of microsporidia, giving priority to molecular characteristics over those observed at the developmental, structural or ultrastructural levels [6]. However, this proposed systematic position has not yet been widely approved by researchers, so in this publication, we use the generally accepted generic name, i.e., *Nosema*.

Honeybee microsporidia attack the midgut cells of bees, where they multiply and produce infectious spores. Additionally, spores present in the middle intestine can cover the

whole intestine lumen [7]. The middle intestine of a honeybee is the focal point of nutrient absorption [8]. Moreover, the symbiotic microflora of the honeybee digestive tract can be also affected by nosemosis infection [9]. Therefore, the production of digestive enzymes and the proper absorption of food compounds in the middle intestine are disturbed during the infection [10,11], which can lead to honeybee death and the depletion of the whole colony. Still, many aspects of this microsporidial disease are unknown; therefore, the aim of this study was to determine the impact of the course of infection on the viability of honeybee intestine cells using propidium iodide (PI) and Syto 9 stains.

## 2. Materials and Methods

One-day-old *Apis mellifera* L. bees were obtained by placing frames with a breaking brood deriving from one queen bee in an incubator at the temperature of 30 °C and an air humidity of 70% [12]. The experiment was carried out on 160 worker bees that were kept in wooden cages settled down by 40 specimens. Worker honeybees were fed with 50% ($w/v$) sucrose solution. On the 3rd day after emergence, the honeybees were divided into two groups: a control group (cages 1 and 2), in which honeybees were left uninfected, and an experimental group (cages 3 and 4), in which honeybees were infected by administering sugar syrup with purified *N. ceranae* spores. The spores were purified to an 85% purity level via the centrifugation of the spore suspension at 5000 G for 5 minutes to produce a pellet of spores. After that, the supernatant was discarded and the pelleted spores were resuspend in distilled water via vortexing for 5 seconds. The whole procedure was repeated 3 times to create an *N. ceranae* spore suspension with an 85% purity level [12]. Honeybees were individually fed with freshly purified 50,000 *N. ceranae* spores/μL in 2 μL of 50% ($w/v$) sucrose solution. Uninfected controls were fed with 2 μL of sucrose solution without spores. Honeybees that had not consumed the entire inoculum were excluded from the experiment [12,13]. of the number of *N. ceranae* spores was estimated using a hemocytometer and an Olympus BX61 light microscope (Olympus Corporation, (Tokyo, Japan)) [12].

The total DNA from the uninfected and *N. ceranae*-infected *A. mellifera* was isolated from the pooled whole abdomens of 3 bees using the DNeasy Blood and Tissue Kit (Qiagen) according to the manufacturer's instructions. The 100 μL of honeybee homogenate was added to 180 μL of lysis buffer and 20 μL of proteinase K, and the total DNA was isolated using the DNeasy Blood and Tissue Kit (Qiagen) according to the manufacturer's instructions. Every isolate was used as a template for the detection of *N. apis* and *N. ceranae*-specific 16S rDNA by performing a PCR test with specific primers: 321-APIS for *N. apis* (5′-GGGGGCATGTCTTTGACGTACTATGTA-3′, 5′-GGGGGGCGTTTAAAATGTGAAACAACTATG-3′) and 218-MITOC for *N. ceranae* (5′-CGGCGACGATGTGATATGAAAATATTAA-3′, 5′-CCCGGTCATTCTCAAACAAAAAACCG-3′) [12,14].

To establish the relative quantification of *N. cearanae* DNA, a real-time PCR was applied according to earlier studies [15–17]. The analysis was completed using the specific 16SSU rDNA region for the *N. ceranae* and *Apis mellifera* β-actin primer pairs, and utilizing the Biometra TOptical Thermal Cycler (Biometra GmbH 846-070-000, TProfessional Basis, Göttingen, Germany). The levels of nucleic acid in *N. cearanae* were quantified in all samples and each well in the PCR plate was loaded with 50 ng of DNA. Standard positive curves were used, and no template controls were run with each plate. Standard curve quantification was used to convert the resulting cycle threshold (CT) values to the number of copies of *N. ceranae* present in each sample. Serial dilutions using purified PCR products with known concentrations from $1 \times 10^9$ to $1 \times 10^2$ copies/μL were used to prepare the standard curve. The products of the amplifications were separated on 2% agarose gel to exclude the presence of primer–dimer structures and non-specific products. The correlation coefficient values were calculated for each sample to ensure the repeatability of the amplicons. The analyses were performed in tree repetitions and in triplicate technical runs. The copy number was expressed as the average *N. cearanae* copy number per bee.

After 21 days, the intestines isolated from 40 healthy and 40 infected bees were prepared. From each intestine, the middle parts of the midgut were isolated for further

analysis (ventriculus recognized using [12]). The isolated parts were stained using two dyes: SYTO 9 and propidium iodide [17]. After staining for 10 min in the dark, excess dye was twice rinsed off using PBS (phosphate-buffered saline). Two microscopic preparates were obtained from each intestine. Live and dead intestine cells were counted from 5 viewpoints of each preparate. In total, 800 data were obtained on the viability of the intestine cells for intestines isolated from healthy and infected bees. The quantification of the cells' viability was completed using an Axiovert 200M fluorescence microscope, Zeiss (Zeiss, Pliening, Germany), immediately after the intestines were staining. Green-colored cells were counted as live, whereas red-colored cells and green cells with a red nucleus were considered dead. The ImageJ 2.1.4.7 i1 program (National Institutes of Health, Bethesda, MD, USA) was used for the quantitative analysis of the cell structure parameters and size. The images obtained were analyzed using morphological transformation sequences, which are available in the ImageJ program. Automatic analyses of images using a microscope were used to determine the percentage of living cells in the images obtained in the studies. The parameters were obtained for a flat section [18–20].

SYTO 9 and propidium iodide stains can also dye *Nosema* spores [17,21], but they are much smaller than intestine cells; *N. ceranae* spores are 3.3–5.5 μm in length and 2.3–3.0 μm in width [1,4], while the diameter of the intestinal cells is approximately 15–20 μm. Therefore, spores can be easily distinguished from honeybee intestine cells in the images. Only dead and live intestinal cells were counted and only intestine cell numbers were included in the statistical analysis.

Both the one-way ANOVA (factor-feeding group) and the NIR post hoc test ($p \leq 0.05$) were applied to estimate the case effects and the significance of the differences between the groups ($\alpha = 0.05$). The Kolmogorov–Smirnov test with the Lileforse correction Statistica software (version 12.0, StatSoft Inc., Tulsa, OK, USA) was used for statistical analyses.

## 3. Results

In the control group, the honeybees remained uninfected, whereas in the experimental group, the infestation level of the honeybees was $15.17 \times 10^6$ ($\pm 2.16$) *N. cearane* copy number/bee. In the samples isolated form infected honeybees, *N. ceranae* spores were visible as characteristic oval shapes (Supplementary Materials Figure S1).

*Nosema ceranae* infection increases from the anterior to the posterior epithelium [13], which is why only the middle parts of the midgut were isolated for the analysis. The viability of cells can be observed using propidium iodide (PI) and Syto 9 stains. PI is a red fluorescent nucleic acid stain and enters only the damaged cells because it is a membrane-impermeant dye that is generally excluded from viable cells. It binds to double-stranded DNA by intercalating between base pairs. PI is excited at 488 and can be used in combination with other fluorochromes excited at 488 nm, such as the SYTO 9 stain, which has an excitation maximum at 483 nm and a fluorescence emission maximum at 503 nm. SYTO 9 stain is an excellent green fluorescent nuclear and chromosome counterstain that is permeant to both prokaryotic and eukaryotic cell membranes [22,23].

After staining, the intestines of healthy honeybees contained more green (live) cells than samples isolated from infected honeybees. When cells appear green fluorescent, this indicates that the membrane is intact and the cell is alive. Red fluorescence indicates the presence of damaged cells, which were observed in a higher amount in *N. ceranae*-infected honeybees. The PI red fluorescent nucleic acid stain only enters the damaged cells, causing a reduction in the SYTO 9 stain because of its higher affinity to nucleic acid. The intestines isolated from *N. ceranae*-infected honeybees on the 21st day contained dead cells concentrated in red/dead spots (Figure 1). The formation of such red/dead spots can lead to necrotic changes and even to an interruption of the intestinal continuity.

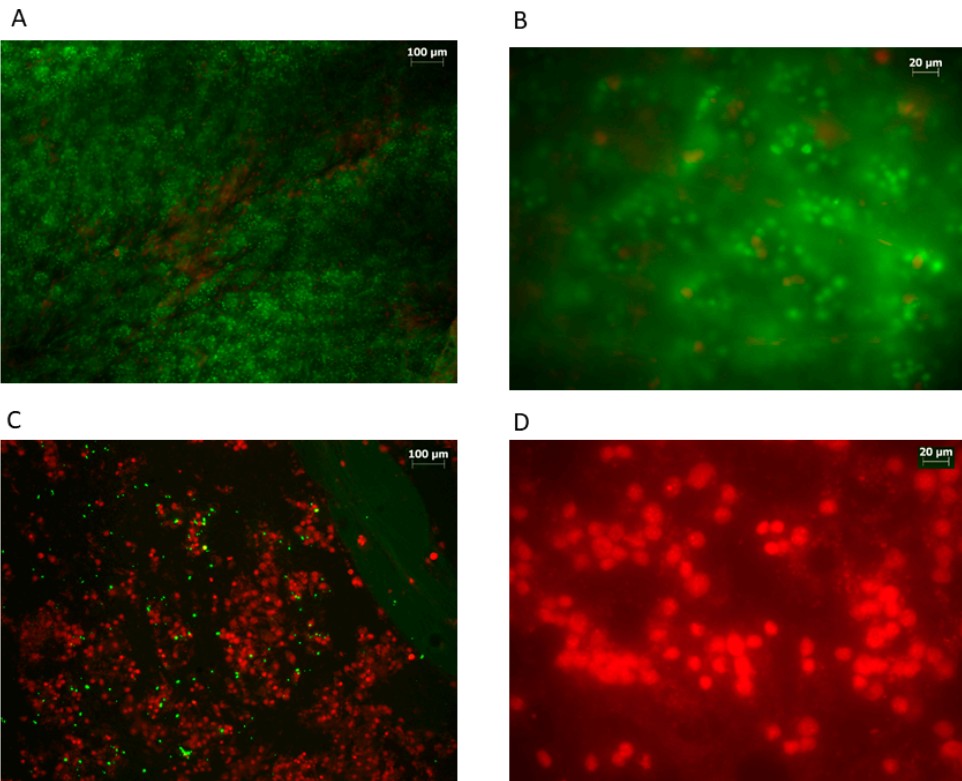

**Figure 1.** Example images of intestines isolated from healthy (**A**,**B**)—magnification 100×, and *N. ceranae*-infected honeybees (**C**,**D**)—magnification 400×. The intestines were stained with live/dead (Syto 9/PI) stains and observed under an Axiovert 200M fluorescence microscope, Zeiss.

All obtained images were the subject of computer image analysis, which is a method of processing and analyzing selected images stored in a digital system. In image analysis, three basic components can be distinguished: (1) image processing, (2) measurements and (3) the interpretation (analysis) of the results. Image analysis programs have many procedures (algorithms) for transforming images. The main four groups of transformations are as follows: geometric transformations (translations, rotations, reflections, distortions), point transformations (points are modified regardless of their proximity), filters (points are modified depending on their surroundings) and morphological transformations (selected points whose surroundings are modified corresponding to the previously modified pattern). Transformations allow the performance of complex operations related to particle analysis. They consist of removing unnecessary details, carrying out measurements on a generalized image, or enhancing certain image elements that are poorly visible. The most important transformations are as follows: erosion, dilation, and opening and closing. Therefore, even images that are blurred for the observer, as in Figure 1B,D, allow quantitative results (the number of live/dead cells) to be obtained after morphological transformations and the use of image analysis.

After the analysis of the obtained images, the number of living cells for healthy and infected bees was obtained. The number of live and dead cells in infected and healthy bees met the requirements of normal distribution for the Kolmogorov–Smirnov test with the Lileforse correction (Figure 2). Based on the results of the significance test and the value of the obtained probability $p < 0.01$ ($n = 800$), it can be concluded that the difference in the number of live and dead intestinal cells of healthy and infected bees is statistically significant.

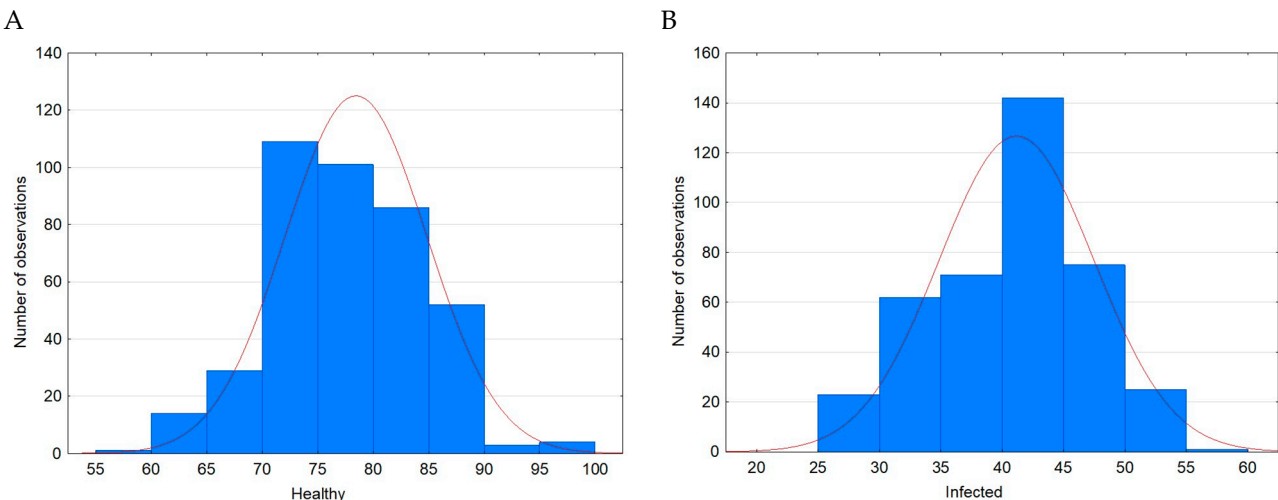

**Figure 2.** The number of live cell observations in the given ranges of the tested number of live cells for healthy (**A**) and infected (**B**) honeybees. The X-axis shows the intervals related to the designated number of cells obtained in each examined image.

The decision regarding normality was made on the basis of a graph in which the values of the analyzed data set are represented in the form of points that are arranged approximately on a straight line, without a clear curvilinear tendency, which is a sufficient argument for the normality of the distribution of the obtained results (Supplementary Materials Figure S2).

Figure 3 presents the graph showing the obtained average values of the number of living cells, with marked standard errors and standard deviation for the tested cases of healthy and infected bees.

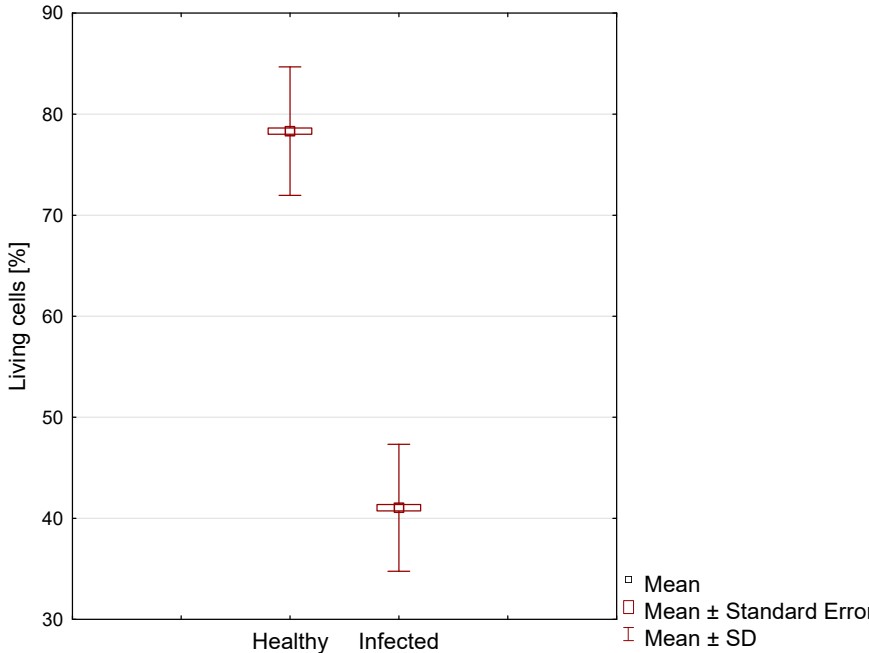

**Figure 3.** Average values of the number of living cells with marked standard errors and standard deviation for the tested cases of healthy and infected bees.

The normality of the distribution is also evidenced by the values of skewness and kurtosis obtained from the statistical analysis, which are close to zero (Table 1). Table 1 presents the results of the calculations and statistical analysis for the examined cases of healthy and infected bees.

**Table 1.** Results of calculations and statistical analysis for the tested cases: healthy and infected honeybees.

| Case | Minimum Value [%] | Maximum Value [%] | Mean [%] | SD [%] | Variance | Skewness | Kurtosis |
|------|-------------------|-------------------|----------|--------|----------|----------|----------|
| Healthy | 59.0 | 96.0 | 78.32 [a] | 6.37 | 40.52 | 0.10 | −0.11 |
| Infected | 26.0 | 56.0 | 41.05 [b] | 6.28 | 39.49 | −0.16 | −0.58 |

SD—standard deviation, different letters (a,b) indicate the occurrence of significant differences between the obtained results.

The tests and analyses performed allowed the number of parameters describing the examined cases to be determined. Maximum values, minimum mean values, standard deviations, the variance in the dispersion of scores around the mean, skewness and kurtosis were determined for the living cells in the tested cases of healthy and infected bees. Healthy bees had a higher number of live cells in their intestines than infected bees. The results obtained for these two cases differed significantly, which was confirmed by statistical tests.

## 4. Discussion

Honeybees can be infected by three species of the *Nosema* genus, i.e., *Nosema apis*, *N. cearanae* and *N. neumanii* [3–5]. *Nosema apis* was described in 1909 by the German biologist Zander [3], *N. ceranae* was originally detected in *Apis cerana* in Asia in 1996 and again recently in *A. mellifera* [4], and *N. neumanni* has presently only been observed in Uganda, but at a higher level than each of the other two *Nosema* species [5]. Nevertheless, the course of these microsporidial infections is similar notwithstanding the species of the pathogens causing the infection. Pathogens from the *Nosema* genus have a high tropism for the ventricular epithelium cells [24]. After the spores' ingestion, a spore polar tube punctures the epithelial cells and infectious sporoplasm enters the host's midgut [7,25]. Next, the pathogen engages the substances and energy in the host's cells to promote its own proliferation, forming a greater number of spores, which are finally released into the intestine's lumen. The release of the spores can be connected with the cell burst of the infected intestine. At the same time, *N. ceranae* inhibits host cell apoptosis to its own advantage, as this process is able to reduce the intensity of the infection [13,26]. Furthermore, *N. ceranae* infection negatively affects the intestine epithelium renewal rate and disrupts the signaling pathways responsible for maintaining homeostasis [27]. All these processes are the reason that microsporidian infection significantly increases the mortality of honeybee workers [7,9–14,17,24,26,27].

*Nosema apis* and *N. ceranae* species were found to infect epithelial cells and clusters of regenerative cells in the honeybee midgut [24]. In this study, we indicated for the first time that the course of infection can lead to the formation of dead intestine cell spots, which can interrupt the host's intestinal continuity and cause severe damage to the host's intestine. This interruption of intestinal continuity is not connected with apoptotic changes, as nosemosis inhibits apoptosis [13], but is most probably associated with the necrosis of ventricular epithelial cells, as has been indicated in earlier studies [28]. This damage to the host's intestine during severe infection may explain why some studies report the presence of pathogenic DNA, not only in the intestine, where it multiplies, but also in other tissues, such as the hypopharyngeal glands, salivary glands, Malpighian tubules, and body fat [16,29]. When spores enter the body cavity they can be transferred by the hemolymph, which flows freely inside the insect body and is in direct contact with organs and tissues. The presence of spores in glands and tissues can disturb their functions and increase the mortality rate of individual honeybees, as well as the whole honeybee colony. For example, the presence of spores in hypopharyngeal glands, which produce and secrete the components of royal jelly for brood and queen feeding, weakens the development and survival rate of the whole colony [29]. The presence of spores in the body fat [16], especially in winter bees that live from 4 to 9 months until spring, can heighten winter losses and reduce the chance of the infected honeybee colony surviving the winter. In conclusion, in

this study, we have demonstrated for the first time that a heavy microsporidian infection can lead to the interruption of the host's intestinal continuity via cellular necrosis, which further contributes to an increase in host mortality.

## 5. Conclusions

Microsporidiosis causes the formation of dead intestine cell spots, which can lead to necrotic changes in the intestine of infected honeybees. This can provoke the interruption of the intestinal continuity, intestinal leaking and an increase in host mortality.

**Supplementary Materials:** The following supporting information can be downloaded at: https://www.mdpi.com/article/10.3390/app13084957/s1. Figure S1. Example images of intestines isolated from healthy (A) and *N. ceranae* infected honeybees (B,C). Figure S2. Diagrams of the normality of the distribution of the number of living cells for the cases of healthy (A) and infected (B) honeybees.

**Author Contributions:** Conceptualization and methodology, A.A.P.; software, M.G.; validation, A.A.P. and M.G.; formal analysis, M.G.; investigation, A.A.P.; resources, A.A.P. and M.G.; data curation, A.A.P. and M.G.; writing—original draft preparation, A.A.P. and M.G.; writing—review and editing, A.A.P. and M.G.; visualization, A.A.P. and M.G.; supervision, A.A.P.; project administration, A.A.P.; funding acquisition, A.A.P. and M.G. All authors have read and agreed to the published version of the manuscript.

**Funding:** This research received no external funding.

**Institutional Review Board Statement:** Not applicable.

**Informed Consent Statement:** Not applicable.

**Data Availability Statement:** All the data are available in the text or in the Supplementary materials. Photos are available at Department of Immunobiology, Institute of Biological Sciences, Faculty of Biology and Biotechnology.

**Conflicts of Interest:** The authors declare no conflict of interest.

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
