# Peer review of "Microsporidiosis Causing Necrotic Changes in the Honeybee Intestine"

_applsci, doi:10.3390/app13084957_

Round 1

Reviewer 1 Report

In the submitted manuscript the authors present a study on the staining of Nosema infected honey bee midgut tissues. The authors show that midgut tissues of Nosema ceranae infected bees have a high proportion of dead epithelial cells compared to uninfected cells.

However, the results are not convincing. The method is not sufficiently described. The method to distinguish between dead and living cells in infected bees was not proved by showing infected epithelial tissue. Furthermore, there are many other issues referring to the methodologies and the study design which have to be addressed.

It is not Vairimorpha. In the publication “A formal redefinition of the genera Nosema and Vairimorpha (Microsporidia: Nosematidae) and reassignment of species based on molecular phylogenetics” the authors redefined the two genera Nosema and Vairimorpha, giving priority to molecular character only. But redefinition based on molecular data is not sufficient. The redefinition is not satisfactory and it just results in some more questions. Additionally, the JIP is not a taxonomic journal.  Redefinitions of species have to be forwarded and determined by taxonomic experts of a taxonomic journal.

Line 48: How where spores purified? How did the authors quantify the spores and what was the dosis for infection?

Line 57-62: What was used as standard for absolute quantification of Nosema ceranae DNA?  

Line 64: Why were Syto 9 and propidium iodide used? What are the references? Propidium iodide is not permeant to living cells. SYTO 9 is permeant to eukaryotic cell membranes and stains DNA. Therefore, the green fluorescence should also be visible in the Nosema infected cells.

Figure 1: There are no microsporidian infections visible in the figure. The authors should proof successful infections by convincing figures. The midgut epithelial cells are not visible. It is not clear which tissue is presented. There are no scales in figure parts C and D. The magnification value is not given. Figure parts B and D are blurry or cloudy. These figures need to be improved.

It is also known that infected cells progressively decline from the anterior towards posterior regions of the midgut in Nosema-infected bees (DOI: 10.1016/j.jip.2018.03.008). What areas of midgut tissues were analyzed by the authors? Was the same part/area analyzed for infected and not infected bees?  

Line 67: view points

Figure 2: What parameter is shown in the x-axes?

Line 93: What are the values for live and dead cells in infected and uninfected bees? What are the values for the standard deviations?

Figure 2: the scales of A and B need to be adjusted to the same size.

Figure 4: The box plots and the standard deviations in the boxplots have exactly the same size. Please check if this is correct. According to figure 1 B and D the number of dead cells in uninfected bees seems to be 0% (B) and in infected bees almost 100% (D). This is not shown/presented in figure 4 or the table 1.

Line 131-147: This is no discussion.

Line 154: Why necrotic cells? The authors did not test for necrotic cells. This is speculative. The entire conclusion was not proved by the experiments.

Author Response

Thank you very much for your insightful remarks. Thank you for your time and effort. We made our best to adhere to your suggestions.

Reviewer 1

In the submitted manuscript the authors present a study on the staining of Nosema infected honey bee midgut tissues. The authors show that midgut tissues of Nosema ceranae infected bees have a high proportion of dead epithelial cells compared to uninfected cells.

However, the results are not convincing. The method is not sufficiently described. The method to distinguish between dead and living cells in infected bees was not proved by showing infected epithelial tissue. Furthermore, there are many other issues referring to the methodologies and the study design which have to be addressed.

Thank you very much for your insightful remarks. Thank you for your time and effort. We made our best to adhere to your suggestions. We have added more detailed methodology concerning the process of sample preparation and in Supplementary materials we have added the images of preparates observed using Olympus BX61 light microscope with

Figure S1. Example images of intestines isolated from healthy (A) and N. ceranae infected honeybees (B, C). The intestines were ob-served under Olympus BX61 light microscope with 10 000X (A, B) and 40 000X (C) magnifications. In the figure C – N. ceranae spores are visible as oval shapes.

It is not Vairimorpha. In the publication “A formal redefinition of the genera Nosema and Vairimorpha (Microsporidia: Nosematidae) and reassignment of species based on molecular phylogenetics” the authors redefined the two genera Nosema and Vairimorpha, giving priority to molecular character only. But redefinition based on molecular data is not sufficient. The redefinition is not satisfactory and it just results in some more questions. Additionally, the JIP is not a taxonomic journal.  Redefinitions of species have to be forwarded and determined by taxonomic experts of a taxonomic journal.

Thank you very much for your insightful remarks. We have corrected the information in the Introduction, lines:37-42

Lastly the taxonomy of Nosematidae family was revised by Tokarev et al. [6], who postulated that Nosema genus should be included into Vairimorpha genus in one monophyletic lineage of Microsporidia, giving priority to molecular characterisctics over those observed at the developmental, structural or ultrastructural levels [6]. However, the proposed systematic position is not yet widely approved by researchers, so in this publication we use the generally accepted generic name, i.e. Nosema.

Line 48: How where spores purified? How did the authors quantify the spores and what was the dosis for infection?

Thank you very much for your insightful remarks. We have added the proper description in Material and Methods, lines: 62-70

The spores were purified to 85% purity level by centrifugation of spore suspension at 5,000 G for five minutes to produce a pellet of spores. After that supernatant was discarded and the pelleted spores were resuspend in distilled water by vortexing for five seconds. The whole procedure was repeated 3 times to create a 85% purity level of N. ceranae spore suspension [12]. Honeybees were individually fed with freshly purified 50,000 N. ceranae spores/µl in 2 µl of 50% (w/v) sucrose solution. Uninfected controls were fed with 2 µl sucrose solution without spores. From the experiment were excluded honeybees that had not consumed the entire inoculum [12,13]. Estimation of N. ceranae spores was accomplished using a haemocytometer and Olympus BX61 light microscope [12].

Line 57-62: What was used as standard for absolute quantification of Nosema ceranae DNA?  

Thank you very much for your insightful remarks. We have added the proper description in Material and Methods, lines: 81-96

To established the relative quantification of N. cearanae DNA a real-time PCR was applied according to earlier studies [15,16,17]. The analysis was completed with specific for the 16SSU rDNA region for N. ceranae and Apis mellifera β-actin primer pairs with Biometra The TOptical Thermal Cycler using primer pairs specific for the 16SSU rDNA (region for N. ceranae and Apis mellifera β-actin. The nucleic acid of N. cearanae levels were quantified in all samples and each well in the PCR plate was loaded with 50 ng of DNA, all samples were run in triplicate technical runs. Standard curves and no template controls were run with each plate. Standard curve quantification was used to convert the resulting cycle threshold (CT) values to the number of copies of N. ceranae present in each sample. Serial dilutions using purified PCR products with known concentrations from 1 × 109 to 1 × 102 copies/μL were used to prepare the standard curve. Positive and no template controls were run with each plate. The products of the amplifications were separated on 2% agarose gel to exclude the presence of primer-dimer structures and non-specific products. The coefficient of correlation values was calculated for each sample to ensure the repeatability of amplicons. The analysis were performed in tree repetitions. The copy number was expressed as the average N. ceranae copy number per bee.

Line 64: Why were Syto 9 and propidium iodide used? What are the references? Propidium iodide is not permeant to living cells. SYTO 9 is permeant to eukaryotic cell membranes and stains DNA. Therefore, the green fluorescence should also be visible in the Nosema infected cells.

Thank you very much for your insightful remarks. We have added the proper description in Material and Methods, lines: 97-100

After 21 days, the intestines isolated from 40 healthy and 40 infected bees were prepared. Form each intestine the middle parts of the midgut were isolated to further analysis (ventriculus recognized using [12]).

And in Results, Lines:122-131

Nosema ceranae infection increases from the anterior to the posterior epithelium [13] that why only middle parts of midgut were isolated for the analysis. Viability of cells can be observed using propidium iodide (PI) and Syto 9 stains. PI is red fluorescent nucleic acid stain and enters only the damaged cells because is membrane impermeant dye that is generally excluded from viable cells. It binds to double stranded DNA by intercalating between base pairs. PI is excited at 488 and can be used in combination with other fluorochromes excited at 488 nm such as SYTO 9 stain with an excitation maximum at 483 nm and fluorescence emission maximum at 503 nm. SYTO 9 stain is an excellent green-fluorescent nuclear and chromosome counterstain that is permeant to both prokaryotic and eukaryotic cell membranes [21,22].

Figure 1: There are no microsporidian infections visible in the figure. The authors should proof successful infections by convincing figures. The midgut epithelial cells are not visible. It is not clear which tissue is presented. There are no scales in figure parts C and D. The magnification value is not given. Figure parts B and D are blurry or cloudy. These figures need to be improved.

Thank you very much for your insightful remarks. Thank you for your time and effort. We made our best to adhere to your suggestions. We have added more detailed methodology concerning the process of sample preparation and in Supplementary materials we have added the images of preparates observed using Olympus BX61 light microscope with

Figure S1. Example images of intestines isolated from healthy (A) and N. ceranae infected honeybees (B, C). The intestines were ob-served under Olympus BX61 light microscope with 10 000X (A, B) and 40 000X (C) magnifications. In the figure C – N. ceranae spores are visible as oval shapes.

We have added the magnification and information below:

  • Figure 1. Example images of intestines isolated from healthy (A,B - magnification 100X) and N. cer-anae infected honeybees (C,D - magnification 400X). The intestines were stained with live/dead (Syto 9/PI) stains and observed under an Axiovert 200M fluorescence microscope, Zeiss.

Results, Lines: 146-159

All obtained images were the subject of computer image analysis, which is a method of pro-cessing and analyzing selected images stored in a digital system. In image analysis, three basic components can be distinguished: 1) image processing, 2) measurements and 3) interpretation (analysis) of the results. Image analysis programs have a lot of procedures (algorithms) for trans-forming images. The main 4 groups of trans-formations are: geometric transformations (translations, rotations, reflections, distortions), point transformations (modify points regardless of their proximity), filters (modify points depending on their surroundings) and morphological transformations (modify selected points whose surroundings correspond to the previously modified pat-tern). Transformations allow to perform complex operations related to particle analysis. They con-sist in removing unnecessary details and carrying out measurements on a generalized image or in enhancing certain image elements that are poorly visible. The most important transformations are: erosion, dilation and opening and closing. Therefore, images, even blurred for the observer as Fig. 1B and Fig.1D, after morpho-logical transformations and the use of image analysis, enabled to obtain quantitative results (the number of live/dead cells).

It is also known that infected cells progressively decline from the anterior towards posterior regions of the midgut in Nosema-infected bees (DOI: 10.1016/j.jip.2018.03.008). What areas of midgut tissues were analyzed by the authors? Was the same part/area analyzed for infected and not infected bees?  

Thank you very much for your insightful remarks. We have added the proper description in Material and Methods, lines: 97-100

After 21 days, the intestines isolated from 40 healthy and 40 infected bees were prepared. Form each intestine the middle parts of the midgut were isolated to further analysis (ventriculus recognized using [12]).

and in Results, Lines:122-131

Nosema ceranae infection increases from the anterior to the posterior epithelium [13] that why only middle parts of midgut were isolated for the analysis.

Line 67: view points

Thank you very much – it was corrected.

Figure 2: What parameter is shown in the x-axes?

Thank you very much. The histogram is used to present the distribution of empirical features, which means that with its help, we present the results we have obtained for certain quantitative variables. It answers the questions (graphically) at what values most of our results are located. Is some value (a range around it) most represented in our dataset? Is there an asymmetry in the results? The histogram is most often used by the researcher to determine the "nature, character of the distribution of the variable". The X-axis shows the intervals related to the designated number of cells obtained in each examined image.

Line 93: What are the values for live and dead cells in infected and uninfected bees? What are the values for the standard deviations?

The values were presented in the table 1.

Table 1. Results of calculations and statistical analysis for the tested cases: healthy and infected honeybees.

Case

Minimum Value [%]

Maximum Value [%]

Mean [%]

SD [%]

Variance

Skewness

Kurtosis

Healthy

59.0

96.0

78.32a

6.37

40.52

0.10

-0.11

Infected

26.0

56.0

41.05b

6.28

39.49

-0.16

-0.58

Figure 2: the scales of A and B need to be adjusted to the same size.

The scale of the Y axis has been unified. In turn, the X scale expresses the intervals related to the designated number of cells obtained in each examined image.

Figure 4: The box plots and the standard deviations in the boxplots have exactly the same size. Please check if this is correct. According to figure 1 B and D the number of dead cells in uninfected bees seems to be 0% (B) and in infected bees almost 100% (D). This is not shown/presented in figure 4 or the table 1.

Thank you very much. As a result of the analysis, similar values of the standard deviation of both examined cases were obtained, and therefore the graph shows similar results. These data are also included in Table 1. The number of living cells for healthy and diseased bees was determined in the study.

Line 131-147: This is no discussion.

Thank you very much. This fragment was moved to the Introduction, lines: 33-42.

Line 154: Why necrotic cells? The authors did not test for necrotic cells. This is speculative. The entire conclusion was not proved by the experiments.

Thank you very much for your comment. We have added the explanation in Discussion, lines: 218-221

This interruption of intestinal continuity is not connected with apoptotic changes as nosemosis inhibits apoptosis [13] but most probably with necrosis of ventricular epithelial cells as was indicated in earlier studies [27].

Thank you very much once more,

Authors

Reviewer 2 Report

Review:

Communication

Microsporidiosis Causing Necrotic Changes in the Honeybee  Intestine

Summary

In the herein presented manuscript “Microsporidiosis Causing Necrotic Changes in the Honeybee Intestine” PtaszyÅ„ska and Gancarz report the effect of a Nosema infection on intestinal cells of honeybees using fluorescent counterstain SYTO 9 and propidium iodide. They report differences in the distribution of live and death cells between uninfected and infected honeybee intestines. The authors conclude that nosemosis in honeybees led to increased death of intestinal cells by quantifying live versus death cell counts.

General comments

The scientific question and conducted experiments were sound, however the manuscript needs major revision before considered for publication.  In addition, I would suggest editing help from someone with full proficiency in English. The introduction is not sufficient to understand the intention of the author’s experiments. References to state the tropism and effects (e.g. Higes et al., 2020; https://doi.org/10.1177/0300985819864302) are missing. The Material and Methods section has to be restructured with subheadings and -sections for bee rearing, PCR, qRT-PCR, dissection and staining, statistics.

The results displayed in the figures are in the current state not acceptable. Which part of the intestine do we see in Fig. 1? I have the impression, that the green fluorescent particles in Fig. 1 D resemble more nosema spores from their size. One thing that has to be included are size bars for Fig. 1C and D. For Fig.2 it is not clear to me, what the x-axis values indicate. In general, I find the information value of Fig. 2, 3 and 4 quite similar. I would recommend to place Fig. 2 and 3 in the supplementary section, and leave Fig. 4 in the main body of the manuscript. The result section needs major rewriting, the figure legends and the result text are from their content very similar. As example, for Fig. 4 the mean percentage between healthy and infected bees should be stated in the text (78% to 41%).

The discussion part is not fulfilling its purpose to discuss the achieved results in the context of known literature, and possible future studies. Some questions that come to my mind are: what is the nature of the differences in live vs. death cells? Is it apoptosis or necrosis? How could this be addressed? Is the counterstaining not also working on nosema spores (see Peng et al., 2014; DOI: 10.1002/cyto.a.22428)? Main effects of nosema infection should be discussed, such as midgut integrity, energy metabolism and immune response. I come to the conclusion that this manuscript should undergo major revisions and rewriting before considering acceptance for publication in MDPI applied sciences.

Specific comments (sorted by line numbering)

13 what justifies the notion that microsporidiosis can lead to intestinal leaking in animal, and in particular human with this study?

45 please specify sugar-water syrup

48 please specify concentration of spores used for infection

55 Add the sequence of the used primers

65 numbers under ten should be written out, so two instead of 2

67 typo, view instead of viuw

68 …In total, 800 data sets were obtained…

70 green colored cells…red colored cells…

72 which version of imageJ

90 stain instead of stein

100 what are the given ranges?

Author Response

Thank you very much for your insightful remarks. Thank you for your time and effort. We made our best to adhere to your suggestions.

Reviewer 2

Microsporidiosis Causing Necrotic Changes in the Honeybee  Intestine

Summary

In the herein presented manuscript “Microsporidiosis Causing Necrotic Changes in the Honeybee Intestine” PtaszyÅ„ska and Gancarz report the effect of a Nosema infection on intestinal cells of honeybees using fluorescent counterstain SYTO 9 and propidium iodide. They report differences in the distribution of live and death cells between uninfected and infected honeybee intestines. The authors conclude that nosemosis in honeybees led to increased death of intestinal cells by quantifying live versus death cell counts.

General comments

The scientific question and conducted experiments were sound, however the manuscript needs major revision before considered for publication.  In addition, I would suggest editing help from someone with full proficiency in English. The introduction is not sufficient to understand the intention of the author’s experiments. References to state the tropism and effects (e.g. Higes et al., 2020; https://doi.org/10.1177/0300985819864302) are missing. The Material and Methods section has to be restructured with subheadings and -sections for bee rearing, PCR, qRT-PCR, dissection and staining, statistics.

Thank you very much for your insightful remarks. Thank you for your time and effort. We made our best to adhere to your suggestions. Thank you very much for pointing very interesting publication. We have added more detailed in Discussion:

Lines: 206-207

Pathogens from Nosema genus have a high tropism for the ventricular epithelium cells [24].

Lines: 218-219

Nosema apis and N. ceranae species were found to infect epithelial cells and clusters of regenerative cells in the honeybee midgut [24].

and Material and Methods, lines: 71-96

Total DNA from uninfected and N. ceranae-infected A. mellifera was isolated using the DNeasy Blood & Tissue Kit (Qiagen) according to the manufacturer’s instruction from pooled whole abdomens of 3 bees. The 100 μl of honeybees homogenates was added to 180 μl of lysis buffer and 20 μl of proteinase K, and the total DNA was isolated by the DNeasy Blood and Tissue Kit (Qiagen) according to the producer’s instruction. Every isolate was used as a template for detection of N. apis and N. ceranae specific 16S rDNA by PCR with specific primers: 321-APIS for N. apis (5’-GGGGGCATGTCTTTGACGTACTATGTA-3’, 5’-GGGGGGCGTTTAAAATGTGAAACAACTATG-3’) and 218-MITOC for N. ceranae (5’-CGGCGACGATGTGATATGAAAATATTAA-3’, 5’-CCCGGTCATTCTCAAACAAAAAACCG-3’) [12,14].

To established the relative quantification of N. cearanae DNA a real-time PCR was applied according to earlier studies [15,16,17]. The analysis was completed with specific for the 16SSU rDNA region for N. ceranae and Apis mellifera β-actin primer pairs with Biometra The TOptical Thermal Cycler using primer pairs specific for the 16SSU rDNA (region for N. ceranae and Apis mellifera β-actin. The nucleic acid of N. cearanae levels were quantified in all samples and each well in the PCR plate was loaded with 50 ng of DNA, all samples were run in triplicate technical runs. Standard curves and no tem-plate controls were run with each plate. Standard curve quantification was used to convert the resulting cycle threshold (CT) values to the number of copies of N. ceranae present in each sample. Serial dilutions using purified PCR products with known con-centrations from 1 × 109 to 1 × 102 copies/μL were used to prepare the standard curve. Positive and no template controls were run with each plate. The products of the am-plifications were separated on 2% agarose gel to exclude the presence of primer-dimer structures and non-specific products. The coefficient of correlation values was calcu-lated for each sample to ensure the repeatability of amplicons. The analysis were per-formed in tree repetitions. The copy number was expressed as the average N. ceranae copy number per bee.

and Results, Lines: 148-162

All obtained images were the subject of computer image analysis, which is a method of pro-cessing and analyzing selected images stored in a digital system. In image analysis, three basic components can be distinguished: 1) image processing, 2) measurements and 3) interpretation (analysis) of the results. Image analysis programs have a lot of procedures (algorithms) for trans-forming images. The main 4 groups of trans-formations are: geometric transformations (translations, rotations, reflections, distortions), point transformations (modify points regardless of their proximity), filters (modify points depending on their surroundings) and morphological transformations (modify selected points whose surroundings correspond to the previously modified pat-tern). Transformations allow to perform complex operations related to particle analysis. They con-sist in removing unnecessary details and carrying out measurements on a generalized image or in enhancing certain image elements that are poorly visible. The most important transformations are: erosion, dilation and opening and closing. Therefore, images, even blurred for the observer as Fig. 1B and Fig.1D, after morpho-logical transformations and the use of image analysis, enabled to obtain quantitative results (the number of live/dead cells).

The results displayed in the figures are in the current state not acceptable. Which part of the intestine do we see in Fig. 1? I have the impression, that the green fluorescent particles in Fig. 1 D resemble more nosema spores from their size. One thing that has to be included are size bars for Fig. 1C and D. For Fig.2 it is not clear to me, what the x-axis values indicate.

Thank you very much for your insightful remarks. We have added propre information in Material and Methods, lines: 98-99

Form each intestine the middle parts of the midgut were isolated to further analysis (ventriculus recognized using [12]).

The spores of Nosema, even the largest ones from N. apis (4-6 μm in length and 2-4 μm in width) are two or three times smaller.

The histogram is used to present the distribution of empirical features, which means that with its help, we present the results we have obtained for certain quantitative variables. It answers the questions (graphically) at what values most of our results are located. Is some value (a range around it) most represented in our dataset? Is there an asymmetry in the results? The histogram is most often used by the researcher to determine the "nature, character of the distribution of the variable". The X-axis shows the intervals related to the designated number of cells obtained in each examined image. The ranges indicate the number of viable cells for which abundance has been specified.

and Results, Lines: 148-162

All obtained images were the subject of computer image analysis, which is a method of pro-cessing and analyzing selected images stored in a digital system. In image analysis, three basic components can be distinguished: 1) image processing, 2) measurements and 3) interpretation (analysis) of the results. Image analysis programs have a lot of procedures (algorithms) for trans-forming images. The main 4 groups of trans-formations are: geometric transformations (translations, rotations, reflections, distortions), point transformations (modify points regardless of their proximity), filters (modify points depending on their surroundings) and morphological transformations (modify selected points whose surroundings correspond to the previously modified pat-tern). Transformations allow to perform complex operations related to particle analysis. They con-sist in removing unnecessary details and carrying out measurements on a generalized image or in enhancing certain image elements that are poorly visible. The most important transformations are: erosion, dilation and opening and closing. Therefore, images, even blurred for the observer as Fig. 1B and Fig.1D, after morpho-logical transformations and the use of image analysis, enabled to obtain quantitative results (the number of live/dead cells).

In general, I find the information value of Fig. 2, 3 and 4 quite similar. I would recommend to place Fig. 2 and 3 in the supplementary section, and leave Fig. 4 in the main body of the manuscript. The result section needs major rewriting, the figure legends and the result text are from their content very similar. As example, for Fig. 4 the mean percentage between healthy and infected bees should be stated in the text (78% to 41%).

Thank you very much, we have placed Fig. 3 in Supplementary materials. Fig. 2 was left in main body of the manuscript and improved according to other reviewer comments.

The discussion part is not fulfilling its purpose to discuss the achieved results in the context of known literature, and possible future studies. Some questions that come to my mind are: what is the nature of the differences in live vs. death cells? Is it apoptosis or necrosis? How could this be addressed? Is the counterstaining not also working on nosema spores (see Peng et al., 2014; DOI: 10.1002/cyto.a.22428)? Main effects of nosema infection should be discussed, such as midgut integrity, energy metabolism and immune response. I come to the conclusion that this manuscript should undergo major revisions and rewriting before considering acceptance for publication in MDPI applied sciences.

Thank you very much for your insightful remarks. Thank you for your time and effort. We changed the Discussion and made our best to adhere to your suggestions. Thank you very much for pointing very interesting publication.

Materials and Methods: Lines 107-109

The ImageJ 2.1.4.7 i1 program (National Institutes of Health, USA) was used for quantitative analysis of the cell structure parameters and size.

Materials and Methods: Lines 132-134

These stains can also dye Nosema spores [17,23] but the spores are much smaller than intestine cells as the biggest spores of N. apis has 4-6 μm in length and 2-4 μm in width [1] and the diameter of the intestinal cells is about 15-20 μm.

Specific comments (sorted by line numbering)

13 what justifies the notion that microsporidiosis can lead to intestinal leaking in animal, and in particular human with this study?

Thank you very much, it was changed to: “Microsporidiosis can lead to the interruption of the intestinal continuity and can lead to intestinal leaking”.

45 please specify sugar-water syrup.

It was changed to “50% (w/v) sucrose solution.”

48 please specify concentration of spores used for infection

Honeybees were individually fed with freshly purified 50,000 N. ceranae spores/µl in 2 µl of 50% (w/v) sucrose solution.

55 Add the sequence of the used primers

Every isolate was used as a template for detection of N. apis and N. ceranae specific 16S rDNA by PCR with specific primers: 321-APIS for N. apis (5’-GGGGGCATGTCTTTGACGTACTATGTA-3’, 5’-GGGGGGCGTTTAAAATGTGAAACAACTATG-3’) and 218-MITOC for N. ceranae (5’-CGGCGACGATGTGATATGAAAATATTAA-3’, 5’-CCCGGTCATTCTCAAACAAAAAACCG-3’) [12,14].

65 numbers under ten should be written out, so two instead of 2

Corrected – thank you very much

67 typo, view instead of viuw

Corrected – thank you very much

68 …In total, 800 data sets were obtained…

Corrected – thank you very much

70 green colored cells…red colored cells…

Corrected – thank you very much

72 which version of imageJ

Versions were used to analyze the images ImageJ 2.1.4.7 i1

90 stain instead of stein

Corrected – thank you very much

100 what are the given ranges?

The ranges indicate the number of viable cells for which abundance has been specified.

Thank you very much once more,

Authors

Round 2

Reviewer 2 Report

Missing points have been addressed, requested changes made, and the text improved by updating references.

Author Response

Thank you very much for your insightful remarks. Thank you for your time and effort. We made our best to adhere to your suggestions.

Reviewer:

Missing points have been addressed, requested changes made, and the text improved by updating references.

Missing points:

  • Are the methods adequately described?
  • Are the conclusions supported by the results?

Thank you very much for your insightful remarks. We have very carefully read the manuscript and corrected the information in the Featured Application, Materials and Methods, Discussion and Conclusions using “Track changes” mode.

We toned down the Featured Application

Lines: 13-15

Microsporidiosis leads to the formation of dead intestine cell spots, which can cause interruption of the host’s intestinal continuity and lead to intestinal leaking.

In Materials and Methods we have removed redundant words and added in lines  119-124

SYTO 9 and propidium iodide stains can also dye Nosema spores [17,21] but the spores are much smaller than intestine cells as spores of N. ceranae are 3.3-5.5 μm in length and 2.3-3.0 μm in width [1,4] and the diameter of the intestinal cells is ap-proximately 15-20 μm. Therefore, spores can be easily distinguished from honeybee intestine cells in the images. Only dead and live intestinal cells were counted and only intestine cell numbers were included in statistical analysis.

We toned down the Conclusions.

lines 251-255

Microsporidiosis caused the formation of dead intestine cell spots, which can lead to necrotic changes in the intestine of infected honeybees. This can provoke interruption of the intestinal continuity, intestinal leaking and an increased host’s mortality.

All text was carefully checked and corrected by English proofreading native speaker.

Thank you very much once more,

Authors